# Effect of Pre-Stress on Laser-Induced Thermoplastic Deformation of Inconel 718 Beams

**DOI:** 10.3390/ma14081847

**Published:** 2021-04-08

**Authors:** Jacek Widłaszewski, Zdzisław Nowak, Piotr Kurp

**Affiliations:** 1Institute of Fundamental Technological Research, Polish Academy of Sciences, Pawińskiego 5b, 02-106 Warsaw, Poland; Jacek.Widlaszewski@ippt.pan.pl (J.W.); znowak@ippt.pan.pl (Z.N.); 2Laser Research Centre, Faculty of Mechatronics and Mechanical Engineering, Kielce University of Technology, al. Tysiąclecia P.P. 7, 25-314 Kielce, Poland

**Keywords:** laser bending, laser-assisted bending, Inconel 718, Johnson–Cook model, curvature

## Abstract

Laser thermal forming is an application of laser heating without any intentional use of external forces. Force-assisted laser bending and laser-assisted bending are hybrid techniques, which combine the use of external forces and local heating to increase the effectiveness of forming. A quantitative description of bending deformation induced by concurrent laser heating and mechanical loading is proposed in this study. Mechanical loading is expressed by the bending moment while the curvature is used to describe the resulting deformation. The contribution of a relatively less known mechanism of laser thermal bending in the hybrid process is identified. The mechanism is able to produce the so-called convex deformation, i.e., bending away from the incident laser beam. Experimental and numerical analysis is performed with thin-walled beams made of Inconel 718 nickel-based superalloy in the factory-annealed state. The Johnson–Cook constitutive material model is used in numerical simulations validated by experimental results.

## 1. Introduction

Simultaneous action of different energy sources in the processing zone can bring the effect of synergy in hybrid manufacturing processes [1,2]. Thermal- and laser-assistance, in particular, have been successfully incorporated into numerous metal forming technologies, e.g., bending, spinning, single point incremental forming (SPIF), roll profiling, stamping, deep drawing, stretch forming, hydroforming and wire drawing [3].

With a growing interest in application of the ultra-high strength steel, high strength aluminum alloys, and such brittle materials like magnesium and titanium alloys, the metal forming processes with local heating of the workpiece have been under development in recent years. Thermal-assistance of metal forming process results in an improved ductility and increased formability of metallic materials at elevated temperatures. Hence, it offers the opportunity to apply smaller forces and pressures, with reduced wear of forming tools, and with the reduction of the spring-back effect [4].

Joule (resistive) heating, which is applied in electrically-assisted forming [5], offers high energy transfer efficiency, but generally involves an increase in temperature of the bulk material, rather than locally, at selected regions where the deformation is needed. Inductive heating is an effective and economically attractive option for relatively large workpieces, e.g., in thermal forming of hull steel plates in shipbuilding [6] or when the workpieces have an axial-symmetric or otherwise regular shape, e.g., for shafts, rods or tubes [7]. In a comparative study on forming with inductive and laser heating, Okman et al. [8] noticed that when the dimensions of the local deformation regions get smaller, the effectiveness of laser heating is increased.

While inductive heating requires a small coil-workpiece distance, laser heating can be performed remotely, thus leaving more space around the workpiece and enabling easier handling and access to it. The availability of localized power sources allows for the development of new processes for metal deformation, in which material flow can be facilitated and controlled at selected regions of desired deformation [8]. Duflou and Aerens [9] reported a 55% reduction of bending force and a similar decrease of the spring-back due to a local preheating of S235JR steel plates. Such reduction of the involved forces is particularly beneficial when bending thick plates or hard materials. Wu et al. [10] analyzed plastic deformation of pre-tensioned strips made of a titanium alloy and heated by the laser beam. The study revealed a large concentration of strain in the heated region.

Application of laser heating offers the ability to introduce local changes of shape of the workpiece in a touchless way. Research on the pure laser thermal forming, i.e., without any intentional application of external forces, started in the 1970s [11,12]. Since then, three fundamental mechanisms were identified [13]: the temperature gradient mechanism (TGM), the upsetting mechanism (UM) and the buckling mechanism (BM). Dependent on the laser processing parameters, part dimensions and topology, different shape changes can be produced by pulsed or continuous localized heating of the material. The bending effect, i.e., the out-of-plane deformation, occurs when: (1) plastic straining results from a steep temperature gradient over the material thickness (the Temperature Gradient Mechanism, TGM), or (2) the heated region undergoes thermal buckling (the Buckling Mechanism, BM). While the instability, inherent to the thermal buckling mechanism, is a limiting factor for its practical use [14,15], the temperature gradient mechanism has a drawback of always producing bends in one direction, i.e., towards the heat source (e.g., the laser beam). Such a bending deformation is termed concave [16] or positive [15] in the literature. The opposite deformation is termed as convex [17] or negative [15]. Influence of external mechanical load on the effect of laser bending was investigated by Vollertsen and Rödle [18] under conditions of TGM. Using a cantilever beam with the dead load on its free end, they observed a decrease of the bend angle with the increase of tensile stress in the laser irradiated surface.

Plastic deformation produced in the pure laser forming is small, which is beneficial for applications in touchless positioning and alignment of components with a micrometer, sub-micrometer or sub-milliradian accuracy [19,20]. However, it is a disadvantage when larger deformations are needed. The reason is the resulting significant time-consumption of the process [21]. In order to increase the forming efficiency, the application of external forces is studied with growing interest [22,23,24,25,26]. In this way, the research on laser forming meets investigations on heat-assisted metal forming processes [27,28].

Laser-assisted bending is a hybrid manufacturing technique for bending plates and other bodies made of metallic and non-metallic materials, including those difficult-to-form ones. Gisario et al. [29] successfully performed bending of Titanium Grade-2 plates, 1 mm thick, with an outside bend radius of 2 mm and bend angles of up to 140°, which was the limit of the mechanical loading device used.

The reported achievements of laser-assisted forming techniques indicate their great potential for shaping difficult-to-form materials or reducing the springback effect. However, the further progress in this field requires a better understanding of the process, as well as the quantitative and predictive description of deformations produced in hybrid forming processes. The complexity of the involved phenomena has so far resulted in a limited understanding of the mechanisms behind the produced deformations. The large variability of the process, the incomplete physical interpretation of the experimental results, the lack of useful analytical models and time-consumption of numerical modeling has directed research efforts towards the methods of statistics and artificial intelligence, such as Artificial Neural Networks, Fuzzy Systems and Genetic Algorithms [26,27,30]. Such a modeling is often aimed at creating a possibly simple black box model, which links main control factors with a measure of processing result.

Yanjin et al. [22] used a metal sheet in a cantilever arrangement, with a mechanical load at the free end. The loading acted so as to bend the sheet toward the laser beam. The research was focused on the effects of material properties on the bend angle. Gisario et al. [27] performed experimental tests aimed at understanding the basic mechanisms involved in the hybrid bending of thin sheets made of AA 6082 T6 aluminum alloy. Laser heating was applied to the constrained samples, which previously had been mechanically deformed beyond the elastic range. In physical interpretation of experimental data, Gisario et al. [27] mentioned two possible mechanisms of the shape correction by laser post-treatment: (1) annealing with the resulting redistribution of the residual stress and (2) laser thermal forming with BM active. Fetene et al. [30] studied the bending behavior of 5052-H32 aluminum alloy sheet, clamped like a cantilever beam with mechanical load on the free end. In an analysis of the bend angle dependence on laser heating parameters and mechanical loading, the researchers tried to use the bending moment as an input parameter. However, drawing clear conclusions from that research was hindered by a significant influence of the shear deformation.

Unfortunately, the contribution of laser thermal forming mechanisms in hybrid bending has not been separated and estimated so far. A fundamental description of the mechanical bending process concerns the dependence of curvature on the bending moment. To the best knowledge of the authors, neither the contribution of laser thermal forming mechanisms to the final deformation, nor the effect of laser heating on the curvature initially produced mechanically, have been quantitatively described for the process of hybrid bending up to now.

The nickel-based superalloy Inconel 718 is one of the most widely applied alloys in the aerospace industry, commonly used in aircraft engines and rocket thrusters for various components of gas turbines, compressor blades, vanes, diffusers, shafts, support cases and other parts. This material is usually provided in the solution annealed condition, but before forming, it can be in the precipitation hardened condition, too. Although Inconel 718 shows good weldability and the general ability to be fabricated in a variety of shapes and forms, it is difficult to form at room temperatures due to its high hardness, work-hardening characteristics and increased tool wear. For example, the high strain hardening of the material has limited its application for multi-sheet cylinder sandwich structures in high-speed vehicles [31]. Fabrication of 3D complex components from Inconel 718 sheet metal is of great importance for the aerospace industry [32,33]. Inconel 718 belongs to the most difficult-to-deformation materials due to its great deformation resistance and narrow hot-working temperature range [34,35]. The herein presented research into hybrid laser-mechanical forming of this superalloy aims at improvements in manufacturing of high-performance components for the aerospace and other industries. The introductory investigation has been performed with X5CrNi18-10 stainless steel [36,37,38].

The objectives of this work are:to determine a material constitutive model of Inconel 718 alloy for numerical simulations of the considered hybrid laser-mechanical forming process;to build a numerical model for the in-depth analysis of the hybrid bending process;to investigate the relation between the local mechanical loading, expressed by the bending moment, and a local change of shape, measured as a change in curvature, while hybrid bending;to identify the role (if any) of laser forming mechanisms in the considered processing;to formulate foundations for the process design.

## 2. Materials and Methods

### 2.1. Experiments

Experimental study of the hybrid bending process was conducted on specimens made of commercial Inconel 718 alloy in the solution annealed condition. The samples tested in the present investigation were laser-cut from a rolled sheet, 1 mm thick, with an average initial grain size of 17.5 μm. Chemical composition of the studied Inconel 718 is shown in Table 1. The 20 mm wide rectangular specimens were clamped in a cantilever arrangement, as shown in Figure 1.

The initial pre-stress condition was realized with the gravity acting on: (1) the mass of the specimen and (2) a set of weights attached to the free end at a distance L = 175 mm from the fixture, as measured along the initially straight specimen. The holder of the weights could freely rotate to provide constantly vertical load force, which is denoted as Q. The following values of the external mechanical load Q (including the weight of the holder and an auxiliary metal plate) were used in a series of experiments: 1.08 N (110 G), 1.57 N (160 G), 2.55 N (260 G), 3.04 N (310 G) and 4.51 N (460 G). The external mechanical load values were chosen so the samples remained in the elastic state while they were initially pre-loaded.

After the specimen had been loaded mechanically, it was heated with a laser beam moving in the direction of axis x, starting from the position x = 150 mm towards the fixed end of sample (x = 0). The laser beam was produced by the TRUMPF TruFlow6000 CO_2_ laser (TRUMPF GmbH + Co. KG, Ditzingen, Germany) (radiation wavelength 10.6 µm), which operated in the continuous-wave (CW) mode. Laser beam of power 500 W had velocity 3.33 mm/s (200 mm/min) with respect to the specimen. The applied laser head with a facet mirror produced an approximately rectangular 20 mm × 2 mm laser spot on the material surface. To present the shape of the laser beam cross-section, Figure 2 shows a view of the surface of a plexiglass plate after the so-called mode shot. In hybrid bending experiments, the laser spot covered the whole width of the specimen.

Surface temperature was measured using OPTRIS CTL G5H CF2 pyrometer (Optris GmbH, Berlin, Germany), which operates at a 5.2 µm radiation wavelength and is insensitive to the CO_2_ laser radiation reflected from the specimen. Optical displacement sensor MicroEpsilon LLT1700 (MICRO-EPSILON MESSTECHNIK GmbH & Co. KG, Ortenburg, Germany) and an auxiliary metal plate (element 4 in Figure 1) were applied to measure the deflection of the specimen.

### 2.2. Numerical Simulations

Numerical simulations of laser-mechanical bending were conducted using the commercial finite element (FE) method program ABAQUS (Version 2016) [39]. The process has been decoupled into two separate analyses. The first one was the heat transfer problem for a beam with a moving heat source (laser beam). This analysis was performed in the initial configuration. The laser beam was modeled as a surface heat source using dedicated user subroutine DFLUX. The homogeneous distribution of the surface power density over the laser spot, i.e., the top-hat profile model, was assumed.

The result of this analysis was a three-dimensional time-dependent temperature field, which was next applied, together with the mechanical load, in the second, quasi-static analysis. Heat generated due to material deformation was assumed to be negligible in comparison to the amount of heat delivered by the laser beam. Dynamic effects from inertial forces were also neglected during the relatively slow deformation process. Hence, calculations were performed in the sequentially coupled analysis, where the material deformation does not change the temperature field [36].

The FE model was developed for a half of the considered beam, because the object, its thermal and mechanical loads and boundary conditions exhibit symmetry with respect to the xz plane (Figure 1a). Three-dimensional linear finite elements with 8 nodes were used: hexahedral DC3D8 elements for thermal problems, and compatible C3D8 elements for mechanical problems. A regular mesh consisted of 141,600 elements of dimensions 0.2 mm × 0.5 mm × 0.1 mm (in directions x, y and z, respectively). Ten layers of elements were used on the beam thickness direction (axis z) in order to accurately model the gradient of the temperature and the bending effect [40]. Finite element mesh used in simulations is shown on Figure 3.

#### 2.2.1. Thermophysical Material Data

Thermal dependences of thermal conductivity, specific heat, linear thermal expansion coefficient and density were taken into consideration (Figure 4 and Figure 5). Temperature-dependent material density was estimated using the formula:(1)$$\rho (T)=\frac{\rho (T_{0})}{1+3\alpha_{T}(T)(T-T_{0})},$$
where: $\rho (T_{0})$ = 8190 kg/m^3^ is the density of Inconel 718 at room temperature, $\alpha_{T}(T)$ is the temperature dependent linear thermal expansion coefficient.

Heat dissipation due to free convection was modeled using the Newton’s law, while the Stefan–Boltzmann’s law was used in describing the radiative heat transfer. Effectiveness of laser beam energy transfer into irradiated material is described by the absorption coefficient (absorptivity), the value of which depends mainly on the radiation wavelength, angle of incidence, material surface quality and temperature. Coupling of energy can be significantly increased by the application of dedicated coatings, which is especially helpful when using the CO_2_ laser beam to heat metallic materials. With its relatively long radiation wavelength, less than 25% of the CO_2_ laser radiation energy is absorbed by the clean surface of Inconel 718, as shown in Figure 6 [43,44].

In order to effectively improve coupling of radiation energy to the material, the samples were coated with a black paint in the presented research. As successful modeling of laser processing vastly depends on the careful estimation of material absorptivity, and there are no theoretical estimations for the particular coating-material combination, the approximate value of the absorption coefficient was determined using an experimental-numerical approach. Surface temperature was recorded during the longitudinal passage of the laser spot across the fixed measurement field of the pyrometer (Figure 7). The strong oscillations seen on the measured data most probably result from burning of the absorptive layer during its irradiation by the laser beam (Figure 1b). The character of this fluctuations suggests that they correspond to the temperature of the absorptive layer and radiation emitted during its burning, rather than changes in temperature of metallic surface underneath. Hence, these oscillations were treated as an artefact. 

Values of parameters describing heat input and heat dissipation, i.e., the absorptivity, convection coefficient and surface emissivity, were estimated in a series of simulations and by the comparison of calculated time-runs of temperature at selected location (x = 113) of material surface with the experimental data (Figure 7). The applied iterative procedure yielded the value of the free convection coefficient to be 5 [W/(m^2^ K)], the surface emissivity of 0.75 and the absorptivity value of 0.37. The same coating and similar processing conditions were used by Kurp et al. [45] for X5CrNi18-10 stainless steel plates heated with the CO_2_ laser. Using an experimental-analytical method, they obtained a close value of the absorption coefficient, namely 0.35.

Absorptivity of metals generally is increased when processing with laser radiation of wavelength shorter than 10.6 μm (Figure 6). A similar heat transfer efficiency as in the considered case of the CO_2_ laser and coating can be obtained using Nd:YAG [43] or diode lasers [41,44] without any coating.

#### 2.2.2. Temperature Field

Figure 8 shows the distribution of temperature for the time instant t = 38 s, and within a selected material region 0 ≤ x ≤ 54. In a reference frame moving with the laser spot, distribution of the highest temperature (above 700 °C) is almost constant on the direction of axis y, with a slight gradient on the direction of material thickness (axis z). The maximal material temperature value does not exceed 800 °C.

Temperature profiles along the axis of the beam are shown in Figure 9. Locations of the top temperature values for time instants t = 1 s, 10 s, 20 s and 40 s correspond to the current locations of the laser spot, while the profile for t = 42 s describes material temperature distribution during cooling.

The maximal material temperature occurs in the laser-irradiated surface (z = 0). According to numerical simulation results shown in Figure 7, the excursion of temperature above 650 °C lasts less than 3 s, and above 700 °C, it is less than 0.5 s. The slow age-hardening behavior of Inconel 718 permits annealing and welding without spontaneous hardening during heating and cooling. Virtually no hardening was observed during the first 2–3 min of exposure to the aging temperature and in the following air cooling [46]. Hence, the short time of laser heating and free cooling are not expected to significantly influence the grain size and to introduce detrimental effects to the material properties.

#### 2.2.3. Material Constitutive Model and Data

An understanding of thermo-mechanical deformation behavior of Inconel 718 within wide range of temperature is a prerequisite for reliable numerical simulation of the considered hybrid bending process. During the last two decades, several constitutive models of Inconel 718 were presented for various quasi-static and dynamic forming processes. Zhang et al. [47] conducted compression tests of Inconel 718 at the temperatures from 960 to 1040 °C, with initial strain rates from 0.001 to 1.0 s^−1^. Many papers are devoted to the finite element simulations of Inconel 718 machining, focusing on cutting force, temperature, chip formation and residual stress, e.g., [48,49,50]. Numerous phenomenological and physically-based constitutive models of nickel-based alloys have been developed in order to understand the interaction of deformation parameters. Among these models, the Johnson–Cook (JC) model [51] and its modifications are extensively used, e.g., [52,53,54]. The JC model is a multiplication typed model which considers strain hardening, strain rate hardening/softening and thermal softening separately.

Typical laser-assisted bending situations involve high temperatures and not so high strain-rates. The regime of processing considered in this research consists of strain-rates from 10^−4^ s^−1^ to 10^−3^ s^−1^ and temperatures up to 800 °C. In order to describe the constitutive response of the as-received Inconel 718 alloy, the following stress $\sigma_{Y}^{T}\left(\varepsilon^{pl},\overset{•}{\varepsilon^{pl}},T\right)$ in the Johnson–Cook model is defined as a function of plastic strain ($\varepsilon^{pl}$), plastic strain rate ($\overset{•}{\varepsilon^{pl}}$) and temperature (T):(2)$$\sigma_{Y}^{T}\left(\varepsilon^{pl},\overset{•}{\varepsilon^{pl}},T\right)=\left[A+B\cdot {\left(\varepsilon^{pl}\right)}^{n}\right]\left[1+C_{JC}\cdot \ln\left(\frac{\overset{•}{\varepsilon^{pl}}}{\overset{•}{\varepsilon_{0}}}\right)\right]\;\left[1-{\left(T^{*}\right)}^{m}\right]$$
where: $\overset{•}{\varepsilon_{0}}$ (0.001 s^−1^ in this work) is the reference strain rate, $T^{*}=\left(T-T_{r}\right)/\left(T_{m}-T_{r}\right)$, $T_{m}$ is the melting temperature (1250 °C in this work) and the transition temperature $T_{r}$ is assumed 20 °C in this work, A is the yield stress under the transition temperature and the reference strain rate, B and n are strain hardening coefficient and exponent, respectively, $C_{JC}$ describes strain rate hardening and m accounts for thermal softening effects. The aforementioned constitutive model is implemented in the ABAQUS finite element program. 

The identification of parameters of the Johnson–Cook model in this research was performed using data from the aforementioned hybrid bending experiments and their numerical simulations. The nonlinear regression approach was applied to find values of parameters A, B, $C_{JC}$, m and n that minimise a functional build on sum of squared differences between experimental and numerical results [55,56].

Numerical simulations consisted of five computational steps describing deformation of the beam due to:step G–gravity acting on the mass of the beam;step Q–external mechanical load Q applied at the free end of the beam;step L–heating with a moving laser beam;step U–unloading due to removal of external mechanical load Q;step F–deactivating gravity (final).

Altogether, seven load cases were considered (see Table 2). They all contain laser heating. The first case (LB–laser bending) concerns the pure thermal load generated by the moving laser beam. In the second case (GL), the laser beam moves along the beam, subject to the gravitational body force acting on its mass. The next five load cases (from Q1 to Q5) take into account effects of the external mechanical load Q applied at the free end of the beam. The loading has been modeled by the vertical surface traction load.

The vertical (*z*-axis) component of displacement vector at the point, where the load Q acts on the beam, was extracted from numerical results and compared to the data measured during experiments (Figure 10). Presented values of deflections were measured from the equilibrium configuration of the beam loaded mechanically. Time was counted since the beginning of laser heating. Simulations were performed for 40 s of laser spot motion from x = 150 mm to x = 16.8 mm. Experimental measurements lasted a few seconds more and included some oscillations of the beam after the laser heating had been finished.

The following material data were assumed at room temperature: Young’s modulus $E_{0}$ = 205 GPa and Poisson’s ratio ν = 0.294. Table 3 presents the values of A, B, $C_{JC}$, m and n obtained from parameter testing and verifying calculation results by the experimental data (Figure 10).

Numerical predictions for all load cases Q1–Q5 are in a good agreement with experimental measurements. Some discrepancies in the obtained results can be attributed mainly to the scatter in the absorptive coating properties between individual samples. The verified numerical model allows for a detailed analysis of the laser-mechanical bending process, which otherwise would be difficult to perform experimentally.

#### 2.2.4. Curvature of the Beam

The combined effect of laser heating and mechanical load was analyzed using the curvature as a convenient quantity, commonly applied in describing bending deformations of beams. The axis of the beam can be treated as a plane curve $\vec{r}=x(s)\vec{i}+z(s)\vec{k}$, where: $\vec{i},\;\vec{k}$ are unit vectors of the axes x and z (Figure 1a), x=x(s) and z=z(s) are parametric equations of the curve and s is the parameter of the curve. It is convenient to use the x-coordinate of the considered beam axis point in the initial (unloaded) configuration as the parameter s. Functions x=x(s) and z=z(s) are available in the discrete form from nodal positions calculated in numerical simulations. The curvature C of the beam axis can be calculated using the equation
(3)$$C=\frac{\left|z^{\prime \prime }x^{\prime }-x^{\prime \prime }z^{\prime }\right|}{{\left({x^{\prime }}^{2}+{z^{\prime }}^{2}\right)}^{3/2}}$$
where operators ()′ and ()″ denote the first and the second derivative with respect to the parameter s, respectively [57]. For simplicity, the curvature and bending moment values are presented with the positive sign in this paper, without formal adherence to the sign conventions. Otherwise, with the assumed coordinate system, all these values should be used and presented in diagrams with the negative sign [58].

The derivatives $x^{\prime }$ and $z^{\prime }$ were computed using the finite difference method. The symmetric difference quotient formula was applied for the first derivative, which for a function *f(s)* is written as
(4)$$f^{\prime }(s)\approx \frac{f(s+h)-f(s-h)}{2h}$$
Similarly, the second derivatives $x^{\prime \prime }$ and $z^{\prime \prime }$ were calculated using the central difference approximation
(5)$$f^{\prime \prime }(s)\approx \frac{f(s+h)-2f(s)+f(s-h)}{h^{2}}$$
Numerical differentiation using a small value of parameter h in the considered problem is sensitive to round-off errors, i.e., to errors due to the limited representation precision of numerical data. The applied value of parameter h in Equations (4) and (5) was 3 mm (for x(s), z(s) and s expressed in millimeters), as this value allowed us to obtain low round-off and truncation errors.

Application of the parametric description of beam configuration ensures adequate treatment of large deflections occurring in the process. The curvature C of the axis was calculated at the end of each calculation step. Curvatures obtained after computational steps *G*, *Q*, *L*, *U* and *F* are denoted as $C_{G}$, $C_{Q}$, $C_{L}$, $C_{U}$ and $C_{F}$, respectively, and completed with the load case number, when needed.

## 3. Results and Discussion

### 3.1. Pure Laser Bending

The final configuration and the curvature of the beam axis for the case of pure laser bending are shown in Figure 11a,b, respectively. The line representing calculated curvature (Figure 11b and the relevant other figures in this paper) shows the following combined effects: (a) the physical effect of deformation induced by laser heating and (b) a numerical effect of the applied derivation procedure, seen as some dense oscillations (‘noise’).

Regarding the former physical effect, three regions of the course of curvature can be distinguished: (1) the location of laser spot start (x = 150) and its neighborhood affected by unstable heating conditions due to the switching the laser beam on, (2) the middle region of stable heating conditions, where the maximum material temperature at the laser spot is practically constant (see Figure 7, x ≤ 131), and (3) the location of the laser spot stop (x = 16.8) and its neighborhood affected by unstable heating conditions due to the switching the laser beam off. The applied numerical derivation procedure based on the finite difference approach also introduces some artefacts at transient regions. Nevertheless, results of performed calculations for the middle region of the beam allow meaningful conclusions and reveal a distinct effect of the pure laser bending. Laser heating produced bending of the beam with the curvature value $C_{LB}$ = 0.797 m^−1^ within the middle region, so the radius of curvature was equal to about 1.25 m there. Bending deformation away from the laser beam, i.e., the so-called convex bending, occurred.

In an attempt to classify the mechanism of the observed deformation, the Fourier similarity number value *Fo* may be used for the major distinction between TGM and other laser forming mechanisms. The Fourier number Fo can be expressed as a relation between the interaction time of the heat source with the material (here $\tau_{h}=d_{h}/v$) and the characteristic time of heat diffusion within the material ($\tau_{h}=H^{2}/\kappa$), $Fo=\tau_{h}/\tau_{d}$, where: $d_{h}$ is the dimension of surface heat source on the direction of velocity v, H is material thickness, κ=λ/(ρ c) is thermal diffusivity, λ is the heat conduction coefficient, ρ is material density and c is specific heat. Hence, for the considered process with a moving heat source, the Fourier number can be expressed as [59,60]:(6)$$Fo=\frac{\kappa \;d_{h}}{v\;H^{2}}$$

For TGM to be active the Fourier number value should be small; Fo ≤ 1. Its optimal value, regarding the maximal magnitude of the bend angle produced under TGM with a laser beam of circular cross-section, is Fo = 0.837 [61].

The approximate value of thermal diffusivity of Inconel 718, at the mean temperature 400 °C of the process considered here, is κ = 4 mm^2^/s. From Equation (6), with $d_{h}$ = 2 mm, the Fourier number value Fo = 2.4 results. Hence, it is evident that laser heating contributes to the considered deformation process with a mechanism different from TGM. The two other fundamental mechanisms BM and UM cannot be directly used to describe the observed character of laser-induced deformation due to the differences in boundary conditions.

A similar behavior of a beam was observed and analyzed in a research on deformations induced by pulsed heating with a stationary laser beam [20,62,63], and with a moving laser beam [64]. The common feature of these processes is the heating of a relatively thin and wide metal beam with a laser spot that covers the whole or a large part of the width of the element. Mucha et al. [62] observed convex bending for Fo > 2. The direction of bending deformation and the process similarity number Fo = 2.4 of the current investigation agree with the findings of Mucha et al. [62].

Considering the direction of the laser-induced bending component in hybrid laser-mechanical processing, the following conclusion can be formulated. The processing should be designed in such a way, that the laser heating would produce bending deformation in the same direction as deformation induced by external forces. Otherwise, an application of two concurring mechanisms would result in a decreased bending efficiency. Consistency of the applied mechanisms of thermal and mechanical forming [65,66] is easy to achieve when TGM is intentionally used. With BM active, the situation becomes more complicated due to the inherent instability involved with this mechanism [27]. The issue can easily be overlooked when the heat source is used just for thermal softening of the material, without taking into account the contribution of thermal forming mechanisms.

### 3.2. Hybrid Laser-Mechanical Bending

Figure 12 presents exemplary diagram of the calculated beam curvatures $C_{G}$, $C_{Q}$, $C_{L}$, $C_{U}$ and $C_{F}$ for the load case Q5. Collected diagrams of curvatures $C_{L}$ and linear approximations of their middle segments are presented in Figure 13a. Similarly, Figure 13b shows diagrams of final curvatures $C_{F}$ for all load cases. Segments between vertical lines define the zone where curvatures $C_{L}$ and $C_{F}$ are changing linearly with the coordinate x.

The magnitude of the bending moment due to the own weight of the beam was calculated as $M_{G}(x)=q{\left(L-x\right)}^{2}/2$, where: q=ρ B H g is the distributed load, B is the beam width, and g is the acceleration due to gravity. Neglecting the influence of large deflections, the total mechanical bending moment can be calculated as $M=M_{G}+M_{Q}$, where the component due to the load Q is assumed $M_{Q}=Q(L-x)$. The approximations of curvatures within the middle segments of linear change are presented in Figure 14 as functions of the total mechanical bending moment M. The relevant curvature-moment data points for all load cases Q1-Q5 were approximated with the following linear functions:(7)$$C_{Q}\left(M\right)=a_{Q}M+b_{Q}\;;\;a_{Q}=2.904\;;\;b_{Q}=0.004\;;\;R=0.999,$$
(8)$$C_{L}\left(M\right)=a_{L}M+b_{L}\;;\;a_{L}=10.33\;;\;b_{L}=0.791\;;\;R=0.998,$$
(9)$$C_{F}\left(M\right)=a_{F}M+b_{F}\;;\;a_{F}=7.379\;;\;b_{F}=0.7795\;;\;R=0.9997,$$
where R stands for the coefficient of determination.

It is clearly seen in Figure 14, that despite all non-linearities involved with the process and the contribution of the pure laser bending mechanism, the curvatures exhibit linear dependence on the bending moment. Some discrepancy of the $C_{L5}$ result may be explained by the effect of neglecting large deflections when calculating the bending moment in the case of the highest mechanical loading (load case Q5). 

The well-known formula for curvature $C_{BE}$ of a beam in pure bending under Bernoulli–Euler assumptions can be written as:(10)$$C_{BE}=a_{BE}M$$
where: $a_{BE}=1/\left(EI\right)$, E stands for Young’s modulus, I is the moment of inertia of the beam cross-section. 

The slope $a_{BE}$ of the analytical curvature-moment characteristics (10) for the beam considered here has the value of 2.92 Nm^−2^. The related slope $a_{Q}$ = 2.904 obtained in calculations (7) differs from $a_{BE}$ by 0.5%. The free term $b_{Q}$ = 0.004 m^−1^ is small and close to the value zero resulting from the analytical dependence for pure bending (10). Laser heating has increased the slope of curvature characteristics by the value $k=a_{L}/a_{Q}$ = 3.56. The free term $b_{L}$ = 0.791 differs from the curvature $C_{LB}$ of the pure laser bending case by about 0.8%. The complete mechanical unloading, both from the force Q and the own weight of the beam, produced a change in the slope of curvature characteristics equal *a_L_* − *a_F_* = 2.947, which differs from the slope *a_Q_* = 2.904 by 1%. The difference in free terms *b_L_* − *b_F_* = 0.01 m^−1^ related to unloading, is close to zero. 

Based on the above observations, the changes in beam curvature for the considered laser-mechanical bending process can be approximately described with the following phenomenological model:(11)$$C_{L}(M)=C_{LB}+k\;C_{Q}(M)$$
(12)$$C_{F}(M)=C_{L}(M)-C_{Q}(M)$$

Equation (11) describes the cumulative effect of: (a) the pure laser bending mechanism and (b) the scaling of the curvature initially produced by the mechanical load. The latter Equation (12) gives a simple estimation of the effect of unloading as a close one to the effect of elastic loading, but taken with the opposite sign.

From Equations (11) and (12), the following formula results:(13)$$C_{Q}(M)=\frac{C_{F}(M)-C_{LB}}{\left(k-1\right)}$$

It can be used to estimate the necessary initial curvature $C_{Q}$ to be induced mechanically in order to obtain the required final (permanent) curvature $C_{F}$ after the process of hybrid laser-mechanical bending. Values of parameters $C_{LB}$ and k should be determined earlier for the actual conditions of processing.

## 4. Conclusions

The effect of pre-stress on the change of curvature in laser-mechanical bending of thin beams was analyzed using numerical simulations. Data from experiments with cantilever beams made of Inconel 718 superalloy, preloaded mechanically in the elastic state and subsequently loaded thermally with a moving CO_2_ laser beam were used to determine parameters of the Johnson–Cook material model applied in the analysis. The developed FE model describes the considered thermoplastic bending behavior in close agreement with results of measurements, which were performed using a broad range of elastic pre-stress level. A phenomenological model for the changes of curvature has been derived. The final curvature after the laser heating step can be estimated by scaling the elastic solution for the mechanically-induced curvature and adding the curvature produced by the pure laser bending. For the effective hybrid bending, the external mechanical load should be applied consistently with the deformation effect of the heat source alone. Under used processing conditions, laser heating without application of any external forces results in convex bending. This effect can be applied in laser bending with the access to the processed element limited to its one side only. The developed predictive model of curvature changes in the considered laser-mechanical bending process can be implemented for the process design, where processing parameters will be optimized.

## Figures and Tables

**Figure 1 materials-14-01847-f001:**
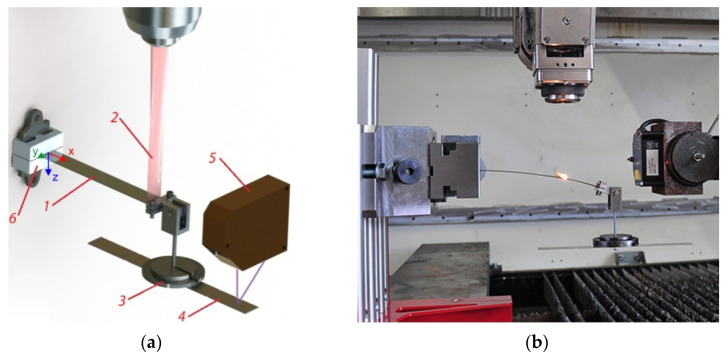
Experimental setup: (**a**) a scheme (1–specimen, 2–laser beam, 3–weights, 4–auxiliary plate, 5–optical displacement sensor, 6–fixture), (**b**) a photograph.

**Figure 2 materials-14-01847-f002:**
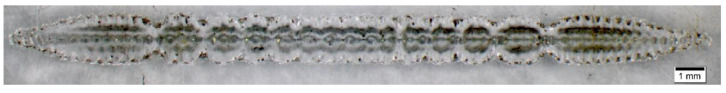
A photograph of a plexiglass plate after the mode shot performed with the applied laser beam.

**Figure 3 materials-14-01847-f003:**
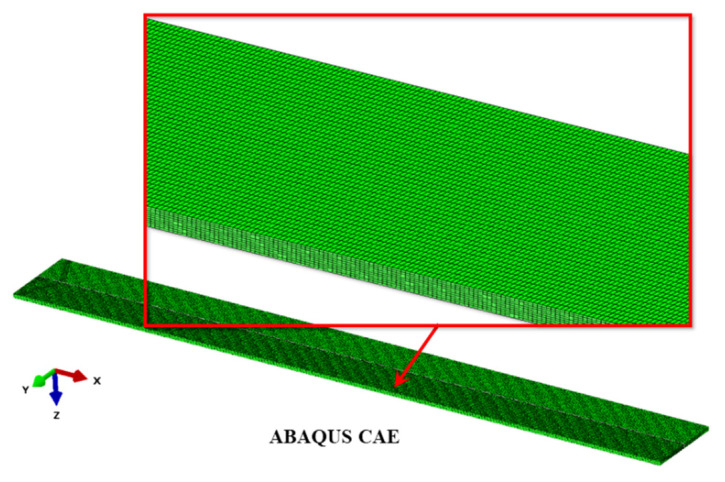
A finite element mesh used in simulations.

**Figure 4 materials-14-01847-f004:**
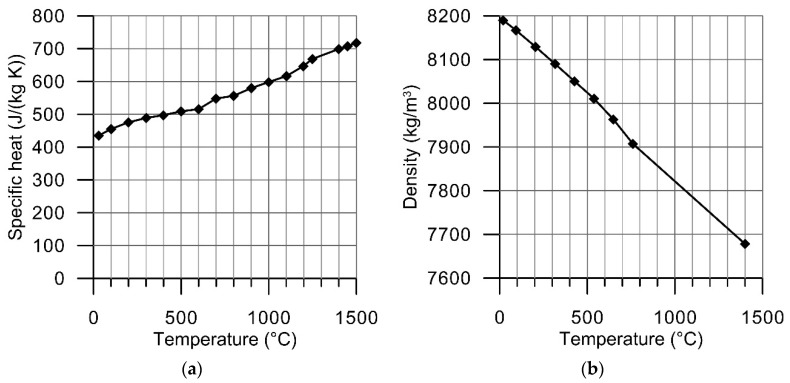
Thermal dependence of: (**a**) specific heat [41], (**b**) density.

**Figure 5 materials-14-01847-f005:**
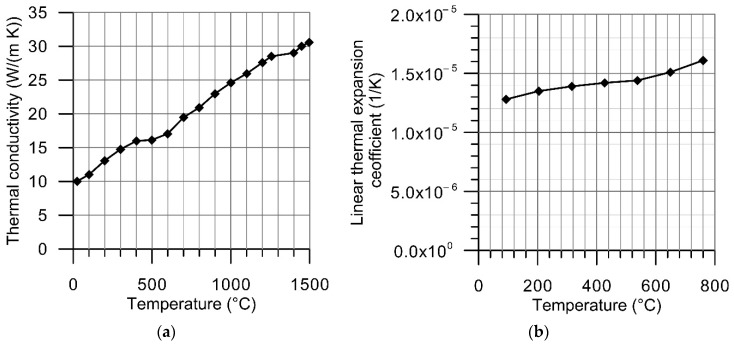
Thermal dependence of: (**a**) thermal conductivity [41], (**b**) linear thermal expansion coefficient [42].

**Figure 6 materials-14-01847-f006:**
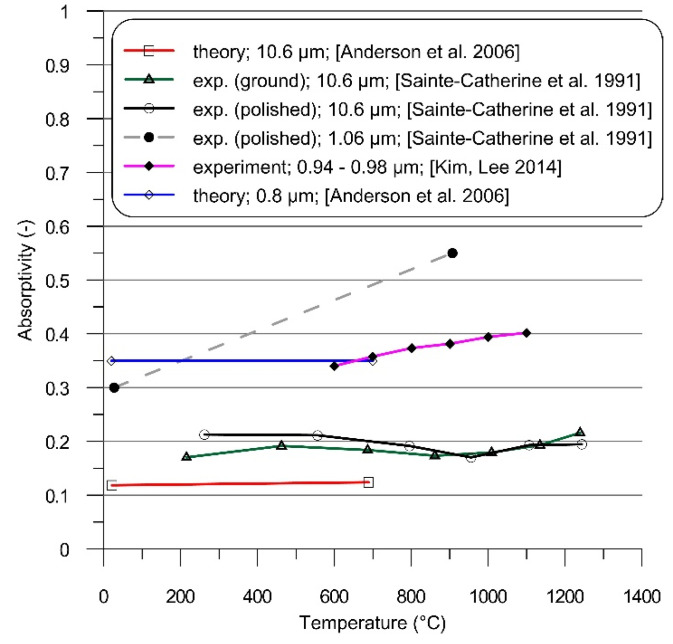
Normal spectral absorptivity of Inconel 718.

**Figure 7 materials-14-01847-f007:**
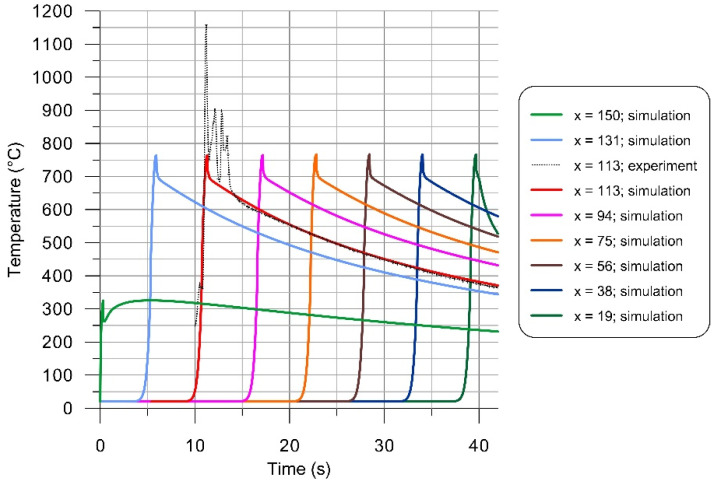
Time-runs of temperature at selected locations x on laser-irradiated surface (y = 0, z = 0).

**Figure 8 materials-14-01847-f008:**
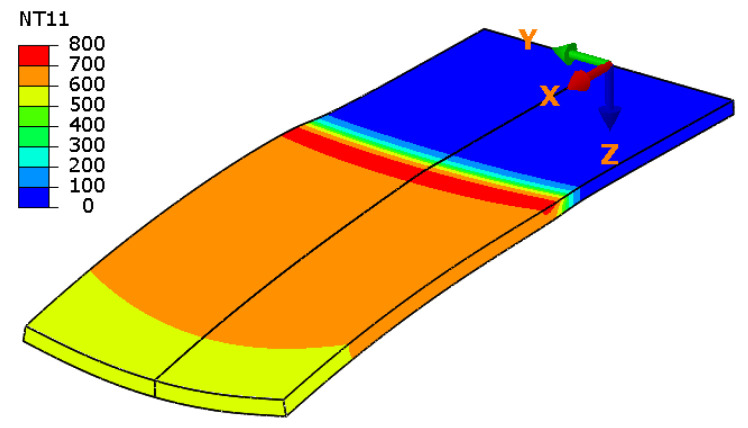
Temperature distribution during laser heating (t = 39 s, region 0 ≤ x ≤ 50, deformation scale factor 10).

**Figure 9 materials-14-01847-f009:**
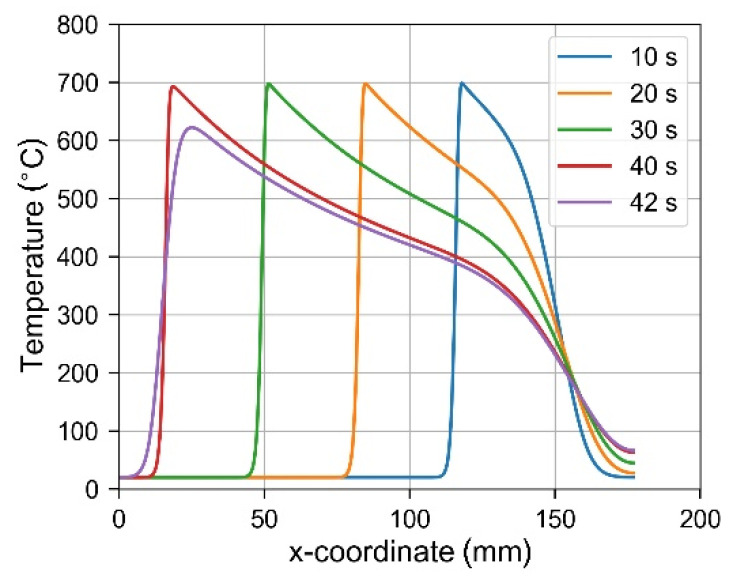
Temperature profiles along the axis of the beam (y = 0, z = 0.5) at selected time instants during laser heating and cooling.

**Figure 10 materials-14-01847-f010:**
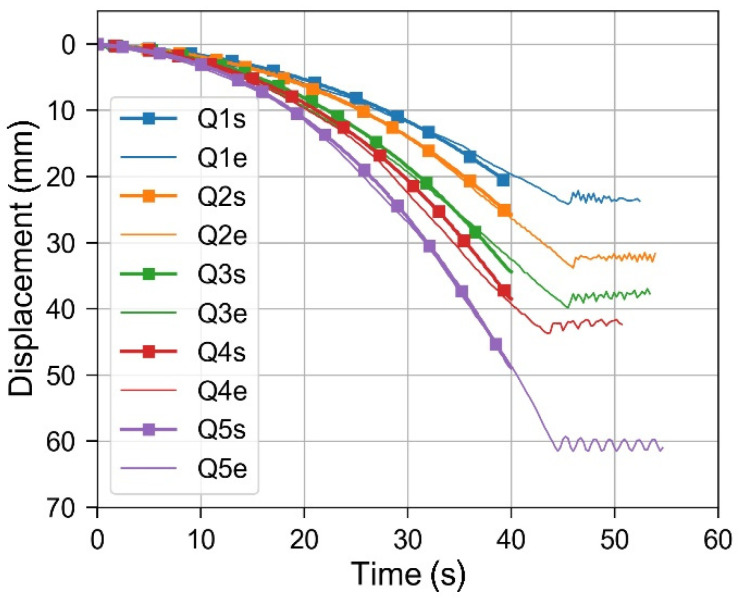
Deflection of the free end of the beam as a function of time for load cases Q1–Q5, measured in experiments (Q1e–Q5e) and calculated in numerical simulations (Q1s–Q5s).

**Figure 11 materials-14-01847-f011:**
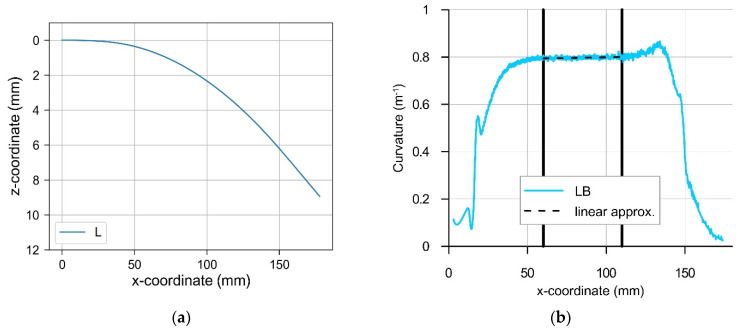
Final configuration (**a**) and curvature (**b**) of the axis of the beam due to the pure laser bending (load case LB, calculation step L).

**Figure 12 materials-14-01847-f012:**
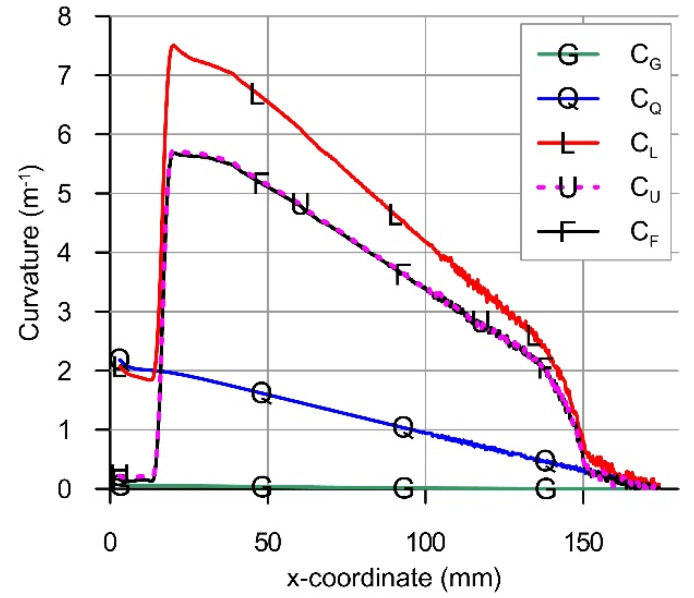
Curvatures $C_{G}$*,*
$C_{Q}$*,*
$C_{L}$*,*
$C_{U}$ and $C_{F}$ of the beam axis in hybrid bending for load case Q5.

**Figure 13 materials-14-01847-f013:**
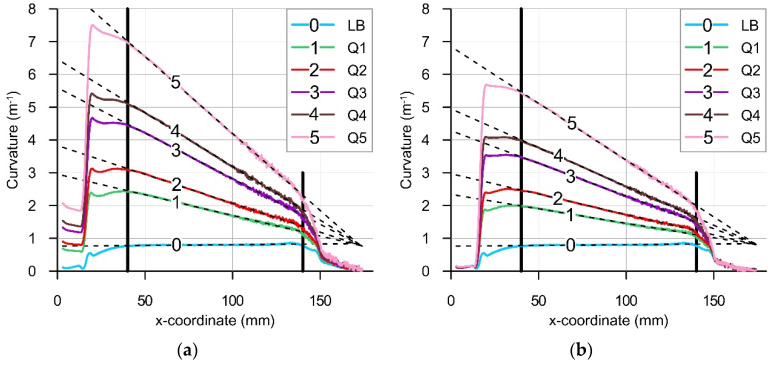
Curvatures of the beam axis in hybrid bending: (**a**) $C_{L}$, (**b**) $C_{F}$.

**Figure 14 materials-14-01847-f014:**
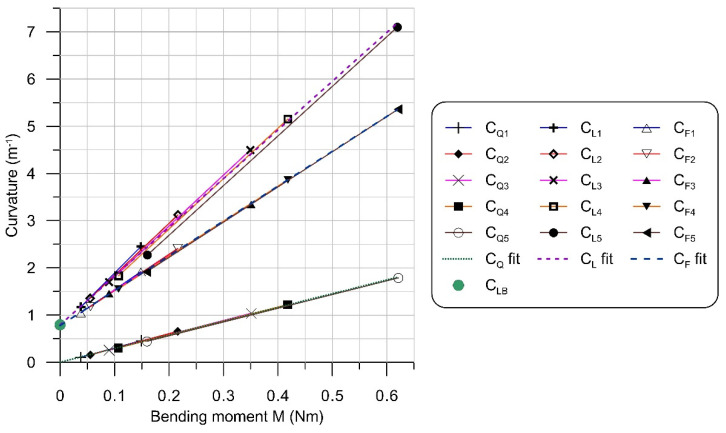
Curvatures after mechanical loading ($C_{Q}$), laser heating ($C_{L}$ ) and complete unloading ($C_{F}$ ), as functions of the bending moment M. Numerical index in the legend denotes the load case number.

**Table 1 materials-14-01847-t001:** Chemical composition of Inconel 718 sheet (wt%).

Ni	Cr	Mo	Nb	Co	Mn	Si	Fe	Al	**Ti**
52.9	19.83	3.12	4.83	0.05	0.29	0.14	Balance	0.60	1.04

**Table 2 materials-14-01847-t002:** List of all load cases and computational steps.

Load Case	Steps				
	G Gravity on	Q Loading Q, N	L Laser	U Unloading Q	F Gravity Off
LB	NO	0	YES	NO	NO
GL	YES	0	YES	NO	YES
Q1	YES	1.08	YES	YES	YES
Q2	YES	1.57	YES	YES	YES
Q3	YES	2.55	YES	YES	YES
Q4	YES	3.04	YES	YES	YES
Q5	YES	4.51	YES	YES	YES

**Table 3 materials-14-01847-t003:** Parameters of the Johnson–Cook (JC) model for Inconel 718 alloy.

A, MPa	B, MPa	n	C_JC_	**m**
450	2100.95	0.76	0.02	1.5

## Data Availability

Data sharing is not applicable to this article.

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
