# Peer review of "Effect of Pre-Stress on Laser-Induced Thermoplastic Deformation of Inconel 718 Beams"

_materials, 2021, doi:10.3390/ma14081847_

Round 1

Reviewer 1 Report

The paper is well prepared and organised. The introduction is complete and it captures the essence of the problem well. The experimental procedure was sufficiently described. Results are clearly shown, and described. Unfortunately, the language used in the paper needs improvement.  Also, some revisions are needed:

Some units should be revised, line 306, 1/s better be s-1; line 300, 1/m better be m-1.

What do the ‘+’ and ‘–‘ represent in Table 2?

More explanation of Eq. (3) should be given.  How was it obtained? A schematic is needed for better illustration.

The figures need to be improved. e.g. Figs. 12,13, use ‘line+symbol’ to differentiate different lines.

Fig. 14, it is difficult to differentiate the lines in the printed black and white version.

Conclusions should be extended by more scientific inference. Currently they are not well summarized and there are too many of them. Most of them can be combined.

Author Response

Thank you very much for Your reviews. All reviews replies were attached to this message. The revised manuscript was attached as well.

Best regards

Authors. 

Reviewer 2 Report

Dear Authors,

I have read your paper "Effect of pre-stress on laser-induced thermoplastic deformation of Inconel 718 beams" carefully. 

This paper describes the hybrid laser-mechanical bending process of Inconel 718 superalloy. The general novelty is the phenomenological model.

The paper is easy to read.

But the methods are not properly described, so that other research groups may not reproduce them.

The paper is interesting. However, it requires few corrections.

  1. Line 64, 65. Please, clarify the acronyms.
  2. Please avoid using we or you, this is not an appropriate style of writing in scientific papers, for example line 305 using you is not good practice. Please check everywhere else in the manuscript and adjust accordingly.
  3. Please, add information about the algorithm of the finite difference method.
  4. Please specifically discuss the advantages of your work. Some parts of your conclusion can be writing in the discussion section. 
  5. References should not be presented in the summary.

 The paper can be accepted for publication only after major improvements.

Author Response

(The authors gave the same response as above.)

Reviewer 3 Report

The paper describes a study of force-assisted laser bending and laser-assisted bending of thin-walled beams made of Inconel 718 nickel-based superalloy. Deformation and thermoplastic deformations process is studied experimentally and numerically by application of the Johnson-Cook constitutive material model. The mechanisms of laser thermal bending are identified. The study is rather limited just to the one size of thin-walled beam. The paper is generally clearly written and well understandable.

Reviewer's comments and remarks to the paper:

1) Please correct missing spaces in the text, e.g. Line 11, 171, 400, 413, and the symbols used, e.g. Line 274, 336.

2) Even well written, the introduction is quite long and could be shortened.

3) The identification of parameters of JC model has to be better described. On which data the model has been fitted? Were the parameters obtained by parametric study? Which range of parameters of model were used for the parametric study? Assuming that all the data in Tab. 2 were used for identification process of the parameters of JC model it is just an calibration of model to the existing results. The application/verification should be then applied to other data, i.e. different loading force or different beam size.

4) In the conclusion no. 5 it's stated that "The verified material constitutive model of Inconel 718...". Such description is inapproprate with regard to remark no. 3 and should be limited just to "An application of the model for description of thermally-assisted laser forming..." Please, correct

5) The study is limited just to thin-walled beam of the one size. Therefore the predictive model of curvature changes should be firstly applied to differents beam sizes or configuration, etc. The point 8 of the conclusion is therefore rather ambitious and should be firstly proven on others configutaions. The authors should consider reformulation of this point.

6) The 9 point of conclusion is Author cotributions. Please, correct the formatting.

Author Response

(The authors gave the same response as above.)

Reviewer 4 Report

The results of the research carried out are devoted to a relevant topic and may be of interest to some readers.
The authors carried out an experimental study and theoretical modeling of Inconel 718 alloy processing processes. The results of the work can be useful to researchers who are engaged in the development of laser methods for metal processing.

The work, in my opinion, can be published after more careful work on the text formatting.
As far as I understand, you need to correct the size of the text fields (left margin)
In table 1, the meaning of the  ">" sign in front of the iron content is not clear. Judging by other concentrations, the iron concentration should be 17.2. Why write "> 17.1"?
In some cases, links to pictures in the test are found after the pictures themselves.

Author Response

(The authors gave the same response as above.)

Round 2

Reviewer 2 Report

Dear Authors,

I have read your modified paper " Effect of pre-stress on laser-induced thermoplastic deformation of Inconel 718 beams " carefully.

Explanations are clear and the paper is easy to read.

I can recommend the Editor to accept this revised manuscript to be published in Materials.